# Perceptual Plasticity in Bilinguals: Language Dominance Reshapes Acoustic Cue Weightings

**DOI:** 10.3390/brainsci15101053

**Published:** 2025-09-27

**Authors:** Annie Tremblay, Hyoju Kim

**Affiliations:** 1Department of Chicano Studies, Languages, and Linguistics, The University of Texas at El Paso, El Paso, TX 79968, USA; 2Department of Psychological and Brain Sciences, The University of Iowa, Iowa City, IA 52242, USA; hyoju-kim@uiowa.edu

**Keywords:** cue-weighting, language dominance, lexical stress, English, Spanish

## Abstract

Background/Objectives: Speech perception is shaped by language experience, with listeners learning to selectively attend to acoustic cues that are informative in their language. This study investigates how language dominance, a proxy for long-term language experience, modulates cue weighting in highly proficient Spanish–English bilinguals’ perception of English lexical stress. Methods: We tested 39 bilinguals with varying dominance profiles and 40 monolingual English speakers in a stress identification task using auditory stimuli that independently manipulated vowel quality, pitch, and duration. Results: Bayesian logistic regression models revealed that, compared to monolinguals, bilinguals relied less on vowel quality and more on pitch and duration, mirroring cue distributions in Spanish versus English. Critically, cue weighting within the bilingual group varied systematically with language dominance: English-dominant bilinguals patterned more like monolingual English listeners, showing increased reliance on vowel quality and decreased reliance on pitch and duration, whereas Spanish-dominant bilinguals retained a cue weighting that was more Spanish-like. Conclusions: These results support experience-based models of speech perception and provide behavioral evidence that bilinguals’ perceptual attention to acoustic cues remains flexible and dynamically responsive to long-term input. These results are in line with a neurobiological account of speech perception in which attentional and representational mechanisms adapt to changes in the input.

## 1. Introduction

The present study examines how language dominance impacts speech perception in highly proficient bilinguals. More specifically, it investigates how bilinguals’ relative experience with their first language (L1) and second language (L2) affects their selective attention to acoustic cues that differ in linguistic informativeness between the two languages (e.g., [1]). Neuroscience research has shown that speech perception is malleable and dynamically responsive to language experience, with cortical systems adapting to changes in input and use over time (e.g., [2,3,4,5,6,7]). The current study builds on this research by examining the effects of language dominance—a proxy for long-term language experience—on the perceptual routines that bilinguals use to process speech in their L2.

Speech perception is a fundamentally multidimensional process: Multiple acoustic cues simultaneously signal linguistic contrasts. For example, as many as 16 acoustic dimensions can characterize the voicing distinction between /ba/ and /pa/ in English [8]. These cues, however, are not perceptually equivalent. Listeners engage in *cue weighting* — selectively attending to some cues more than others based on their informativeness for categorization in a given language (e.g., [9,10,11,12]). Cue weighting is, therefore, language-specific, shaped by a listener’s exposure to the statistical regularities of acoustic realizations in the input. A key concept for understanding how cue weighting is learned and modified is *selective attention*, a cognitive mechanism by which listeners focus on particular acoustic dimensions while down-weighting others (e.g., [9,11]). This process is formalized in attention-to-dimension models of perceptual learning, which posit that experience with a particular language stretches or compresses dimensions in perceptual space, emphasizing contrastive cues and de-emphasizing redundant ones (e.g., [13,14]).

Crucially, bilinguals’ experience with their L1 has a profound impact on how they allocate attention to acoustic cues in other languages (e.g., [1,15,16,17,18,19,20,21,22,23,24,25,26,27,28,29]). Cross-linguistic transfer effects are well documented among *late* L2 learners, who begin learning their L2 after childhood (for a review, see [30]). These age-of-acquisition effects may reflect several factors, including longer experience with the L1 before exposure to the L2 and possible changes in neurocognitive development. Evidence for strict maturational constraints in speech perception, however, is inconclusive; rather, research suggests that age-related changes may be driven by L1 experience and interact with other factors such as L2 proficiency and attention control (e.g., [31]). Importantly, continued L2 experience can modify perceptual attention even in adulthood (e.g., [1]), though less is known about how this occurs in highly proficient bilinguals with more balanced or L2-dominant experience. Perceptual learning research shows that the speech perception system remains highly adaptable: Brief exposure to a shifted category boundary can lead to perceptual retuning (e.g., [32,33,34]), and adaptation to accented speech can occur within minutes (e.g., [35,36]). These findings point to a perception system that is flexible and dynamically responsive to input (see also [37]) but raise the question of whether this adaptability extends to the reorganization of cue weighting in bilinguals (e.g., [38,39,40]), particularly high-proficiency bilinguals who vary in their language dominance.

*Language dominance* refers to a bilingual’s relative proficiency in, exposure to, use of, and accessibility to their two languages; it is shaped by the bilingual’s language history, use, proficiency, and attitudes (e.g., [41,42,43,44]). It is not synonymous with age of acquisition or L1 status, and it may shift over time in response to changes in language exposure and use. It is also influenced by attitudes towards the L1 and L2. Importantly, language dominance has been shown to modulate a wide range of linguistic processes, including morphosyntactic processing (e.g., [45]), lexical access (e.g., [46]), and phonetic categorization (e.g., [47]). Yet, its influence on acoustic cue weighting in speech perception remains understudied, particularly in bilinguals with high proficiency in both languages. Results from Dmitrieva [40] are consistent with the prediction that cue weighting reflects experience-driven perceptual routines: Native Russian speakers residing long-term in the US, the majority of whom were late L2 learners of English, patterned with monolingual English speakers in their cue weighting for English stop voicing contrasts (e.g., /k/ vs. /g/) but diverged from monolingual Russian speakers when performing an equivalent task in Russian. These results suggest that cue weighting is modifiable through L2 experience and lend credence to the hypothesis that language dominance may exert a strong influence on how bilinguals attend to language-specific acoustic cues.

Neurocognitive research further supports the idea that speech perception is highly flexible and continuously shaped by experience. From a neural perspective, this flexibility is underpinned by dynamic patterns of cortical plasticity that enable listeners to adapt to different linguistic demands. For instance, using functional magnetic resonance imaging (fMRI), Myers and Mesite [3] found that sensitivity to a shifted phonetic boundary initially emerged in right frontal and middle temporal regions, suggesting that this initial adjustment may stem from the auditory processing regions. Over time, this sensitivity shifted to left superior temporal areas—regions associated with phonetic categorization—indicating neural adaptation to altered acoustic distribution. In a subsequent multi-voxel pattern analysis of the same data, Luthra et al. [2] showed that a classifier trained on listeners’ brain activation patterns for unambiguous stimuli could predict their responses to ambiguous tokens at above-chance levels. These findings suggest that perceptual categories are neurally encoded and modifiable by recent experience, and that perceptual retuning is reflected in distributed cortical activity (see also [4,6,48]). This supports a neurobiological account of speech perception in which attentional and representational mechanisms adapt to changes in the input, in line with experience-based models of cue weighting.

The present study builds on this perspective by investigating whether language dominance shapes selective attention to acoustic cues in highly proficient bilinguals. If the perceptual system is indeed flexible and experience-driven, as the neurocognitive literature suggests, then the relative weighting of cues in bilingual speech perception should vary within bilingual populations as a function of their dominant language. By focusing on highly proficient bilinguals, this study isolates the influence of long-term language experience from proficiency. Such an approach will help clarify whether the perceptual system can be restructured and whether language dominance can reshape how listeners attend to language-specific acoustic cues. This question is particularly relevant in the domain of lexical stress perception, where languages differ not only in whether they have lexical stress but also in how stress is realized acoustically, and which cues are prioritized by listeners. As such, lexical stress provides a strong test of whether bilinguals who are dominant in the L2 exhibit a cue-weighting profile that differs from bilinguals who are dominant in the L1. In this study, we focus on L1-Spanish L2-English listeners’ perception of English lexical stress.

English uses lexical stress to convey differences in meaning (e.g., *DEsert* vs. *deSSERT*, with capital letters indicating the stressed syllables) or lexical category (e.g., *REcord* [n.] vs. *reCORD* [v.]). Among the various acoustic cues that signal lexical stress, vowel quality is the most important in English: Stressed syllables contain a full (unreduced) vowel (e.g., /ε/ in *DEsert*), whereas unstressed syllables are more likely to contain a reduced vowel (e.g., /ə/ in *deSSERT*) (e.g., [49,50]). Reduced vowels differ from full vowels in both articulatory centralization and shorter duration. Even in the absence of strong vowel reduction (e.g., *PERmit* [n.] vs. *perMIT* [v.]), duration remains a consistent correlate of stress: Stressed syllables are longer than unstressed ones, though this difference is attenuated by phrase-final lengthening (e.g., [51,52]). In contrast, pitch serves primarily to signal intonational pitch accents and phrase boundaries: Depending on its prosodic context, the same word can be produced with a falling contour (e.g., a L* pitch accent at the end of a declarative sentence: *Mary ate all the dessert*), a rising contour (e.g., a H* pitch accent at the end of a yes-no question: *Did Mary eat all the dessert?*), or a flat contour (e.g., no pitch accent following a contrastively accented word: *It was Mary who ate all the dessert*) [53,54,55,56,57]. As a result, pitch can only be interpreted as a cue to lexical stress when sentence-level intonation is taken into account (e.g., [58,59,60,61]). Intensity has also been found to be an acoustic correlate of English lexical stress, but because its effect has been reported to be negligible (e.g., [17,18]), the present study will not investigate the use of this cue. Hence, the use (or non-use) of this cue in English or Spanish will not be discussed.

These cue distributions inform how English listeners perceive lexical stress. Previous research has shown that monolingual English listeners rely much more on vowel quality than on pitch or duration when perceiving lexical stress (e.g., [1,18,19,62,63,64]), and they attend more to pitch than to duration (e.g., [1,17,19]). Although duration is a reliable stress cue in English, listeners appear to make limited use of it, perhaps because it is redundant with vowel quality. Studies on spoken word recognition have also shown that English listeners are more sensitive to lexical stress when unstressed vowels are reduced than when they are full (e.g., [65,66,67]), although suprasegmental cues can still support lexical access in the absence of segmental cues (e.g., [65,68,69,70]).

Like English, Spanish uses lexical stress to convey differences in word meanings (e.g., *PApa* ‘pope’ vs. *paPÁ* ‘daddy’); additionally, stress in Spanish can mark grammatical distinctions such as verb tense and person (e.g., *LLEgo* ‘[I] arrive’ vs. *lleGÓ* ‘[s/he] arrived’ (e.g., [71,72]). Among the acoustic cues that signal lexical stress in Spanish, duration is a consistent correlate: Stressed syllables are longer than unstressed ones across intonational contexts (e.g., [73,74]). As in English, pitch also signals intonational pitch accents and phrase boundaries in Spanish, and it serves as a cue to lexical stress only in the context of sentence-level intonation (e.g., [74,75,76,77]). Crucially, unlike English, Spanish does not use vowel reduction to signal lexical stress, with vowels retaining their full quality in unstressed syllables (e.g., [71,72]). This cross-linguistic difference raises the possibility that, compared to English listeners, Spanish listeners may attend more strongly to pitch and duration cues when perceiving English lexical stress.

Evidence from perception studies supports the prediction that Spanish listeners attend to pitch and duration cues to lexical stress. Research by Dupoux and colleagues (e.g., [78,79,80]) has shown that Spanish listeners are highly sensitive to lexical stress in nonce words where stress is cued solely by suprasegmental information. This sensitivity has also been observed in spoken word recognition (e.g., [81]). When individual acoustic cues are isolated, Spanish listeners have been found to rely more on pitch than on duration (e.g., [82,83,84,85]). However, few studies have examined how Spanish listeners perceive English lexical stress. One study comparing Spanish and Korean L2 learners of English found that Spanish listeners outperformed Korean listeners in an oddity task targeting lexical stress perception (e.g., *conVERT*, *PERmit*, *INcense*, where the oddball is the first word), suggesting that experience with lexical stress in the L1 confers a perceptual advantage; yet, because the stimuli were not controlled for vowel quality, it remains unclear which cues Spanish listeners relied on [86].

To better understand which cues bilingual listeners prioritize and how cue weighting may shift with language experience, the current study examines whether L1-Spanish L2-English bilinguals with varying degrees of language dominance differ in their weighting of vowel quality, pitch, and duration cues to English lexical stress. Prior research on cue weighting in L2 learners’ perception of English lexical stress has revealed robust transfer effects from the L1, leading to the formulation of the *Cue-Weighting Transfer Hypothesis* [1,20,21,22], a hypothesis presumed to apply to the perception of other types of contrast as well (see also [15]). The transfer of cue weightings has been documented both within and across phonological and functional domains—not only from L1 to L2 lexical stress (e.g., [1]), but also from L1 lexical tones or pitch accents to L2 lexical stress (e.g., [16,17,20,62,64,87]), from L1 lexical stress to L2 lexical tones (e.g., [88]), and from L1 lexical stress to L2 speech segmentation (e.g., [22]). In addition, findings from this line of work suggest that it is more difficult for listeners to suppress reliance on a cue that carries greater weight in the L1 than learning to attend to a cue that is less heavily weighted in the L1 but informative in the L2. This asymmetry is evidenced in bilinguals outperforming monolinguals on tasks where the acoustic cue is much more meaningful in the L1 (e.g., [17,27,62]) by proficiency effects emerging only for the latter (e.g., [1]), a pattern echoed in the L2 spoken word recognition literature (e.g., [89,90]).

However, most of the existing cue-weighting research with bilinguals has focused on late L2 learners of English, whose L1 experience far exceeds their L2 experience. Hence, it remains to be seen whether L2 listeners who are dominant in the L2 can demonstrate reduced reliance on cues that are informative in the L1 but not in the L2. Addressing this question is critical not only for testing the limits of the Cue-Weighting Transfer Hypothesis but also for advancing broader theories of bilingual speech perception, which must account for how cue-weighting routines evolve with shifting patterns of language dominance. Additionally, this research is important for broader theories of bilingual language learning and plasticity: Demonstrating that bilinguals can adjust reliance on L1-preferred cues in favor of L2-preferred ones would suggest that the perceptual system remains dynamically reconfigurable well into adulthood, rather than being fixed by early experience. It would also clarify how listeners integrate multiple cues in real-world learning contexts, where speech is rarely processed in isolation from other dimensions. More broadly, such findings would contribute to ongoing debates about the limits of perceptual flexibility in L2 acquisition and provide a framework for distinguishing systematic bilingual adaptations from impaired processing.

Building on this body of research, and considering the cross-linguistic differences in how lexical stress is cued in English and Spanish, the present study was designed with two main aims:To determine whether Spanish–English bilinguals and English monolinguals differ in their reliance on vowel quality, pitch, and duration when perceiving English lexical stress.To test whether language dominance predicts individual differences in bilinguals’ cue weighting, with the expectation that English-dominant bilinguals will show greater reliance on vowel quality and reduced reliance on pitch and duration compared to Spanish-dominant bilinguals.

We predict that cue weighting will vary systematically with language dominance. Specifically, English-dominant bilinguals are expected to rely more on vowel quality than their Spanish-dominant peers, reflecting increased sensitivity to the primary stress cue in English. More critically, if speech perception is flexible and dynamically shaped by language experience, English-dominant bilinguals are predicted to rely less on pitch and duration, which are more informative in Spanish than in English, than Spanish-dominant bilinguals. These predicted patterns would support the view that language dominance, rather than L1 status alone, plays a central role in shaping cue weighting. Such findings would also provide behavioral evidence for plasticity in speech perception, suggesting that the underlying neural systems remain responsive to long-term experience with the L2.

As discussed earlier, language dominance is a multifaceted construct shaped by bilinguals’ language history, use, proficiency, and attitudes [41,42,43,44]. The aim of this study is not to disentangle the individual contributions of these factors or to make claims about the attainability of native-like performance in the L2 after a certain age but to investigate how this construct as a whole modulates L2 speech perception.

## 2. Materials and Methods

The data for this study were collected for Minjárez-Oppenheimer [91], a subset of which were reported in the thesis.

### 2.1. Participants

Thirty-nine L1-Spanish L2-English bilinguals (mean age: 21.3, standard deviation [SD]: 4.6, range: 18–39, 33 female) participated in the study. They were recruited and tested at a large state university on the U.S.-Mexico border. The bilinguals’ language history, use, estimated proficiency, and attitudes were documented using a modified version of the Bilingual Language Profile (BLP; 42 see Appendix A, [92]). Table 1 summarizes their Spanish–English language background.

All bilinguals had learned Spanish before the age of 2, and they began learning English between the ages of 0 and 18 (mean: 6.3 years); accordingly, they reported feeling comfortable using Spanish earlier (1 year) than English (9 years). On average, they received more years of instruction in English (12.1 years) than in Spanish (6.8 years) and had spent more time living in the US (13.7 years) than in Mexico (7.2 years). They reported more frequent use of Spanish with family (67.4%) but more frequent use of English with friends (55.4%) and in school/work settings (71.5%). They also reported greater use of English when talking to themselves (56.2%) and when counting (55.1%). Self-rated proficiency scores were higher in English than in Spanish for listening (5.6 vs. 5.3), speaking (5 vs. 4.6), reading (5.5 vs. 4.6), and writing (5.2 vs. 3.9), with larger gaps between the two languages in written skills. Participants reported feeling like themselves when speaking either language (Spanish: 4.8, English: 4.7), but they identified more strongly with a Spanish-speaking culture (4.8 vs. 3.9 for English), placed greater value on using Spanish like a native speaker (5 vs. 4.7 for English), and were more likely to want to be perceived as native speakers of Spanish. In summary, although the bilinguals learned Spanish first and had more positive attitudes towards it, they had more years of experience with English and rated themselves as more proficient in English.

The BLP questionnaire was modified to include additional questions targeting the participants’ border-commuter experience and bilingual language use (see Appendix A). The questionnaire was modified because some of the questions from the original tool do not reflect the reality of Spanish–English bilinguals who live on the U.S.-Mexico border. For example, participants whose country of residence is Mexico may still spend a significant amount of time in the US if they are border commuters. The questionnaire was modified to reflect this reality. Twelve bilinguals reported currently commuting across the border, with an average of 4.7 years of commuting experience (SD: 3.7, range: 1–13). As a group, participants reported having lived with family members who speak both Spanish and English for an average of 15.8 years (SD: 6.7, range: 0–20) and having attended school or worked in environments where both languages were commonly spoken for an average of 13 years (SD: 6.1, range: 1–20).

Language dominance scores were calculated following the procedures described in Gertken et al. [42], with one modification: The Language History score included only the first four questions from the original BLP (listed in Table 1), as the remaining questions were adapted and no longer relativistic. Language dominance scores ranged from −70.2 (Spanish-dominant) to 43.8 (English-dominant), with a group mean of −20.9 (SD: 29.9), indicating that the group was slightly more Spanish-dominant, though with substantial variability.

To assess English proficiency, the bilinguals also completed a 50-item open-ended “cloze” (i.e., fill-in-the-blank) test [93]. This test was formerly part of the English Language Institute Placement Test at the University of Hawai‘i (J. D. Brown, personal communication, 28 April 2007) and is widely recognized as a valid and reliable measure of global English proficiency, including lexical, morphosyntactic, and discourse knowledge. Responses were scored as correct if they matched a pre-established list of acceptable answers (see Appendix A). The test is considered challenging: English monolinguals score around 44, with a range of 34 to 50 [94]. The bilinguals achieved a mean score of 36.3 (SD: 6.6, range 22–46), indicating overall advanced proficiency in English, though scores varied substantially across individuals. A Pearson correlation between language dominance and cloze test performance did not reveal a significant relationship between the two measures (*r =* 0.05, *p* > 0.1; see Appendix A).

To assess how the bilinguals’ knowledge of Spanish influences their perception of English lexical stress, 48 native English speakers without advanced knowledge of Spanish (henceforth referred to as English monolinguals) were also tested. These participants were recruited and tested at a large state university in the Midwest. After being tested, eight participants were excluded from this control group: Two were simultaneous Spanish–English or English–Hindi bilinguals, and three had been exposed to Spanish, German, or Haitian Creole at home between birth and the age of 5. An additional three participants were excluded due to self-reported advanced proficiency in Spanish. The final native English speaker group included 40 participants (mean age: 21, SD: 4.3, range: 18–40, 28 female) who reported growing up speaking English, being educated primarily in English, and not having advanced knowledge of Spanish.

None of the participants reported having hearing or speech impairments.

### 2.2. Materials

Participants completed the cue-weighting stress-perception task from (Tremblay et al. [1]; see that paper for full methodological details). Auditory stimuli were based on recordings of the word pair *DEsert–deSSERT* produced by a female native speaker of American English in a neutral carrier sentence. A single token of each word was selected for acoustic manipulation. Table 2 (also in [1]) presents the acoustic measurements of these naturally produced tokens.

The stimuli were manipulated in Praat (version 6.0.46; Boersma & Weenink, 2019 [95]) to vary along three acoustic dimensions that signal lexical stress: vowel quality, pitch, and duration. Vowel quality was adjusted by gradually transforming the spectral characteristics (i.e., first, second, and third formant frequencies and bandwidths) of the vowels from *DEsert* to *deSSERT* in seven evenly spaced steps, producing a continuum from one vowel realization to the other. Pitch and duration were similarly varied in seven steps between the two endpoint words. For each stimulus, two dimensions were varied independently while the third was held at a neutral midpoint, resulting in three sets of stimuli: vowel quality × pitch, vowel quality × duration, and pitch × duration. Overall intensity was neutralized across the two syllables and normalized across items.

This design produced a total of 147 unique items (3 matrices × 49 stimuli), each presented across three blocks for a total of 441 test trials. Twelve additional practice trials were included with canonical realizations of stress on either the first or second syllable (i.e., all cues aligned at Step 1 or Step 7). This stimulus set allowed us to examine how bilinguals and monolinguals rely on each acoustic dimension when perceiving English lexical stress.

### 2.3. Procedures

Participants completed the stress perception task after a visual-world eye-tracking task and a sequence-recall task. They sat individually at a computer workstation in a quiet laboratory setting. The experiment was programmed and administered using Gorilla (www.gorilla.sc; [96]). At the start of each trial, participants were prompted to play the auditory stimulus by clicking a play button. After the stimulus played, the following sentence appeared on the screen: *Press 1 for “desert” and 2 for “dessert.”* Once a response was entered, the next trial began with a new prompt to play the next audio file. The task was therefore entirely self-paced.

The experiment began with a brief practice phase consisting of 12 stimuli in which vowel quality, pitch, and duration cues to stress converged; participants received explicit feedback on their accuracy during this practice phase. Participants were required to reach at least 75% accuracy in the practice session to be included in the data analysis. All participants reached this criterion. The main session followed immediately and comprised three blocks in which the 147 test stimuli were fully randomized. The full experiment lasted approximately 20–25 min.

### 2.4. Data Analysis

All analyses and visualizations were conducted in R [97] using the tidyverse [98], brms [99] and ggplot2 [100] packages. We implemented statistical models within a Bayesian framework using brms [99], with models fitting performed via the cmdstanr backend. Bayesian modeling was chosen because it provides full posterior distributions, allowing probabilistic statements about the credibility of effects (e.g., *p* (*β* > 0)), and handles continuous predictors and complex hierarchical structures more robustly than frequentist models, which sometimes exhibited convergence issues. Bayesian methods also naturally accommodate uncertainty in parameter estimates and facilitate visualization of marginal effects with credible intervals, which is especially useful for interpreting interactions with continuous predictors.

Participants’ binary responses (*DEsert* = 1, *deSSERT* = 0) were analyzed as a function of the relative weighting of three acoustic cues: vowel quality, pitch, and duration. The analyses focused on two populations: L1-Spanish L2-English bilinguals differing in their language dominance, and English monolinguals. All acoustic predictors were centered around the mean to facilitate interpretation of main effects and interactions. For bilinguals, standardized language dominance scores (i.e., language-dominance z-scores) were used as a continuous predictor. Item was included as random effect to account for the within-item variation across the three repetitions of the auditory stimuli.

We conducted two sets of analyses: The first compared bilinguals and monolinguals, with the bilingual group as statistical baseline; the second examined the role of language dominance as a continuous variable within the bilingual group. For visualization purposes only, participants were divided into terciles based on their language dominance scores: Spanish-dominant, more balanced, and English-dominant.

Separate logistic regression models were fit for each cue pair (formant × pitch, formant × duration, pitch × duration). For the bilingual-monolingual comparison, the model formula was: Response ~ Cue1 × Cue2 × L1 + (1|Participant) + (1|Item); for the bilingual-only dominance analysis, the structure was: Response ~ Cue1 × Cue2 × LanguageDominance + (1|Participant) + (1|Item). All models assumed a Bernoulli likelihood and used four chains of 4000 iterations each (with 1000 warm-up iterations). Conservative control settings (adapt_delta = 0.95, max_treedepth = 12) were used to ensure stability and convergence.

Posterior distributions of the model parameters were summarized using posterior means, 95% credible intervals, and posterior probabilities for each fixed effect (i.e., *p* (*β* > 0) and *p* (*β* < 0)). Interactions of interest (e.g., cue × L1, cue × language dominance) were probed by generating marginal effects using conditional_effects () and visualizing model-based predictions with 95% credible intervals. Results are considered credible when the posterior probability that an effect is greater (or less) than zero exceeds 0.95.

## 3. Results

### 3.1. Vowel Quality by Pitch

Figure 1 displays participants’ responses to stimuli that varied along the vowel quality and pitch continua. The left panel shows the mean results for Spanish listeners, while the right panel shows those for English listeners (for bilingual listeners’ individual responses, see Appendix A). The *x*-axis represents the 7-step vowel quality continuum, with Step 1 indicating acoustic properties associated with word-initial stress and Step 7 indicating those associated with word-final stress. The *y*-axis reflects the parallel 7-step continuum for pitch, with the same directional mapping. Color shading indicates the proportion of *DEsert* responses: Darker purple reflects more frequent *DEsert* selections, whereas lighter purple indicates more *deSSERT* responses. All stimuli in this figure had a fixed, mid-range value (Step 4) on the duration continuum. Table 3 summarizes the results of the Bayesian logistic regression model for these data, including posterior probabilities for each fixed effect.

The Bayesian model in Table 3 reveals several credible effects. Vowel quality and pitch both had strong negative effects on Spanish listeners’ responses, with lower proportion of *DEsert* selection as vowel quality and pitch steps increased. The effect of L1 was negative, with higher proportion of *DEsert* selection among Spanish listeners than among English listeners across cues. Credible interactions also emerged from the analysis: The interaction between vowel quality and L1 was negative, indicating that Spanish listeners showed a weaker negative effect of vowel quality compared to English listeners. The interaction between pitch and L1 patterned in the opposite direction, revealing that Spanish listeners showed a stronger negative effect of pitch compared to English listeners. The model-based predictions for these two interactions are represented visually in Figure 2. These results indicate that, compared to English listeners, Spanish listeners relied less on vowel quality and more on pitch when perceiving English lexical stress.

A parallel analysis of bilingual listeners’ responses that included language dominance instead of L1 revealed several credible interactions. Recall that, in these analyses, language dominance is entered in the models as a continuous variable, and participants are divided into terciles only for visualization purposes. First, the effect of vowel quality became more negative as language dominance increased, *β* = −0.114, 95% CI [−0.146, −0.083], with a posterior probability of *p* (*β* < 0) >0.999. Second, the effect of pitch became less negative with increasing language dominance, *β* = 0.080, 95% CI [0.050, 0.111], *p* (*β* > 0) > 0.999. Model-based predictions for these two interactions are shown in Figure 3 (for the full model, see Appendix A). Together, these results suggest that language dominance modulated Spanish listeners’ reliance on acoustic cues to English lexical stress, with English-dominant bilinguals patterning more similarly to monolingual English listeners.

### 3.2. Vowel Quality by Duration

Figure 4 shows participants’ mean responses to stimuli that varied in vowel quality and duration (for bilingual listeners’ individual responses, see Appendix A). This time, the *y*-axis represents the 7-step duration continuum, with the same directional mapping as the vowel-quality continuum. All stimuli in this figure had a fixed, mid-range value (Step 4) on the pitch continuum. Table 4 summarizes the results of the corresponding Bayesian logistic regression model and posterior probabilities for each fixed effect.

The Bayesian model in Table 4 shows that both vowel quality and duration had strong negative effects on Spanish listeners’ responses: the proportion of *DEsert* selections decreased as vowel quality and duration steps increased. Several credible interactions were also found. The interaction between vowel quality and duration was positive. Visual inspection suggests that Spanish listeners were less sensitive to vowel quality at the higher end of the duration continuum, indicating that duration cues favoring *deSSERT* may have biased interpretation and reduce reliance on vowel quality. The interaction between vowel quality and L1 was negative, reflecting a weaker vowel quality effect for Spanish listeners compared to English listeners. In contrast, and mirroring the pattern found for pitch, the interaction between duration and L1 was positive, with Spanish listeners showing a stronger negative effect of duration than English listeners. Model-based predictions for these two interactions are shown in Figure 5. Finally, the model revealed a credible three-way interaction between vowel quality, duration, and L1, which was negative. This indicates that the two-way vowel quality × duration interaction observed in Spanish listeners was attenuated in English listeners. Taken together, these results suggest that, compared to English listeners, Spanish listeners relied less on vowel quality and more on duration, and they showed reduced sensitivity to vowel quality when duration cues strongly favored *deSSERT*.

An additional analysis of bilingual listeners’ responses with language dominance instead of L1 also revealed several credible interactions. As with the previous set of stimuli, the effect of vowel quality became more negative as language dominance increased, *β* = −0.104, 95% CI [−0.136, −0.072], with a posterior probability of *p* (*β* < 0) > 0.999. Additionally, the effect of duration became less negative with increasing language dominance, *β* = 0.032, 95% CI [0.001, 0.063], *p* (*β* > 0) > 0.98. Model-based predictions for these two interactions are represented in Figure 6 (for the full model, see Appendix A). These results again indicate that language dominance modulated L1-Spanish L2-English listeners’ reliance on acoustic cues to English lexical stress, with bilinguals who are more dominant in English patterning more like monolingual English listeners.

### 3.3. Pitch by Duration Stimuli

Figure 7 displays participants’ mean responses to stimuli that varied in pitch and duration (for bilingual listeners’ individual responses, see Appendix A). The *x*-axis now represents the 7-step pitch continuum, with the same directional mapping as the duration continuum. All stimuli in this figure had a fixed, mid-range value (Step 4) on the vowel quality continuum. Table 5 summarizes the results of the corresponding Bayesian logistic regression model and posterior probabilities for each fixed effect.

As shown in Table 5, the Bayesian model revealed several credible effects. Pitch and duration had negative effects on Spanish listeners’ responses, with a lower proportion of *DEsert* selection as pitch and duration steps increased. The effect of L1 was again negative, with a higher proportion of *DEsert* selection among Spanish listeners than among English listeners across cues. Credible interactions also emerged: The interactions between pitch and L1 and between duration and L1 were both positive, revealing that Spanish listeners showed stronger negative effects of pitch and duration compared to English listeners. The model-based predictions for these two interactions are represented visually in Figure 8. These results indicate that Spanish listeners relied more on pitch and duration than English listeners did.

A similar analysis of bilingual listeners’ responses but with language dominance instead of L1 revealed two interactions in the same direction, one credible and one marginally credible: The effects of pitch became less negative as language dominance increased, *β* = 0.060, 95% CI [0.039, 0.089], *p* (*β* > 0) > 0.999, and so did the effect of duration, *β* = 0.025, 95% CI [−0.003, 0.054], *p* (*β* > 0) < 0.959. Model-based predictions for these two interactions are shown in Figure 9 (for the full model, see Appendix A). Overall, these results again indicate that language dominance modulated Spanish listeners’ reliance on acoustic cues to English lexical stress, with English-dominant bilinguals reducing their use of pitch and duration cues relative to Spanish-dominant bilinguals.

## 4. Discussion

The present study addressed two main questions: (1) whether Spanish–English bilinguals differ from English monolinguals in their reliance on vowel quality, pitch, and duration when perceiving English lexical stress and (2) whether language dominance predicts individual differences in cue weighting among highly proficient bilinguals. Consistent with our first aim, the results revealed clear cross-linguistic differences: Compared to monolingual English listeners, Spanish–English bilinguals relied less on vowel quality and more on pitch and duration when identifying stress in English words. These differences reflect the influence of L1 experience on cue weighting, supporting the notion that selective attention to acoustic dimensions is language-specific. Critically, in line with our second aim, language dominance emerged as a credible predictor of cue weighting within the bilingual group: Bilinguals with higher English dominance scores patterned more like English monolinguals, showing both increased sensitivity to vowel quality and decreased sensitivity to pitch and duration; in contrast, Spanish-dominant bilinguals exhibited a perceptual profile more aligned with Spanish, giving greater weight to pitch and duration. These patterns held consistently across all cue pairings, indicating that language dominance systematically modulates selective attention to language-specific acoustic dimensions in speech perception. These results should be interpreted as reflecting relative dominance across multiple dimensions rather than a single objective metric, underscoring the multifaceted nature of this construct.

These findings support experience-based models of cue weighting (e.g., [9,10,11]), which posit that selective attention to acoustic cues is not fixed but dynamically shaped by the statistical distribution of cues in the input. The present results extend this body of work by demonstrating that perceptual flexibility is not limited to short-term learning contexts but also reflects long-term, cumulative language experience, with highly proficient bilinguals adjusting their cue-weighting patterns in accordance with their dominant language. This supports the view that speech perception remains flexible and that changes in language dominance can reshape core aspects of perceptual attention, in line with attention-to-dimension models of perceptual learning (e.g., [13,14]). The current findings thus challenge theoretical accounts that treat L1 transfer as fixed [79,80] and instead highlight the importance of dominance as a dynamic and functionally meaningful construct in bilingualism research.

Furthermore, the present findings offer robust behavioral evidence for the Cue-Weighting Transfer Hypothesis (e.g., [1,20,21,22]), which posits that listeners transfer cue-weighting routines from the L1 to the L2. Our findings replicate this transfer pattern among Spanish–English bilinguals as a group and among Spanish-dominant bilinguals, who showed greater reliance on pitch and duration compared to English monolinguals and English-dominant bilinguals. Examining the cues individually, we observed that vowel quality sensitivity increased markedly with English dominance, while pitch and duration sensitivity decreased, highlighting an asymmetric reweighting process consistent with the relative informativeness of each cue in the two languages. Crucially, the results also reveal that L1-Spanish L2-English bilinguals can reduce their reliance on pitch and duration cues as their dominance in English increases. This pattern extends previous findings by demonstrating that high-proficiency, L2-dominant bilinguals can recalibrate their attention even to cues that were initially more salient in the L1. To our knowledge, this is among the first study demonstrating that bilinguals can reduce their attention to cues that are more informative in the L1 than in the L2. Previous research had revealed persistent transfer of such cues in the L2 (e.g., [16,17,20,62]), with the use of such cues not decreasing at a higher L2 proficiency (e.g., [1]). Our results indicate that with sufficient experience in the L2, bilingual listeners readjust their perceptual attention to *all* cues, not just the ones that are more informative in the L2 compared to the L1. These findings support an experience-based interpretation of L1 influence in bilingual speech perception, showing that it is not static or immutable but dynamically shaped by relative language exposure and use (see also [30,101]).

Beyond replicating and extending prior work, the present findings have broader consequences for how we conceptualize language learning and plasticity. The dynamic cue reweighting observed here demonstrates that perceptual attention to prosodic cues is not rigidly constrained by the L1 but instead reflects ongoing adaptation to patterns of input across both languages. This provides behavioral evidence for plasticity in bilingual speech perception, showing that the weighting of prosodic dimensions can be recalibrated with sufficient L2 experience. Although our study examined cues in isolation, the results have clear implications for cue integration: If bilinguals shift their reliance away from pitch and duration and toward vowel quality when processing English stress, similar shifts are likely to shape how they integrate multiple cues in more naturalistic contexts. While the dominance-related differences we observed were relatively small in magnitude, their systematicity suggests genuine underlying perceptual adjustments. Importantly, even subtle shifts in cue weighting can have cascading effects in speech processing, potentially delaying lexical access if more competitors are activated when the cues are not used efficiently. Future work will need to test whether the shifts we observed in the current study extend to complex, multi-cue environments such as continuous speech. The present findings therefore contribute to a broader understanding of bilingual learning, supporting models in which perceptual systems flexibly redistribute attention across cues as a function of language experience. They also underscore that bilingual speech patterns, often described as “non-native”, should not be taken as evidence of impairment but instead as evidence of systematic adaptations to bilingual input [102,103].

These findings also offer indirect support for neurobiological accounts of speech perception that emphasize distributed, experience-sensitive cortical representations. Research has shown that perceptual learning is associated with shifts in cortical activity, particularly in regions involved in cue encoding and categorization (e.g., [2,3,4,6,48]). The pattern of cue reweighting observed here, shaped by long-term language experience, suggests that attentional mechanisms involved in speech perception are capable of reorganizing in response to shifting input distributions. This interpretation aligns with neurocognitive research showing that listeners can rapidly adapt to altered acoustic distributions and that this adaptation is reflected in cortical reorganization. More broadly, these results support the view that bilingual speech perception is both language-specific and experience-contingent. Rather than representing a fixed compromise between two systems, the bilingual perceptual system adapts the distribution of acoustic cues to contrasts, which in turn influences how listeners attend to different cues in the speech signal.

Despite the robustness of these findings, several limitations should be acknowledged. First, the participant pool was relatively small and narrow, consisting only of 39 highly proficient Spanish–English bilinguals, most of whom were female. Our design allowed us to isolate the role of dominance while controlling for proficiency and age of acquisition of Spanish, but it also limits the generalizability of the results to bilinguals with other L1-L2 pairings and possibly to male bilinguals (though gender has not been reported to influence cue weightings in speech perception). Second, language dominance was assessed only through the self-report measures provided by the Bilingual Language Profile [42,92]. Although widely used and informative, such measures may not provide the same degree of precision or accuracy as objective behavioral measures.

Future research should therefore aim to replicate and extend these findings with bilingual populations from different language backgrounds and with complementary measures of dominance. It would also be valuable to examine whether similar dominance-based cue-weighting patterns emerge in bilingual populations from language pairs that differ more dramatically in their prosodic system; for example, where one language is tonal and the other is not. Speakers of tonal languages such as Mandarin are known to transfer their use of pitch to the perception of lexical stress in English, a finding attributed to the high functional load of pitch from lexical tones in the L1 (e.g., [16,17,62,64]). However, it remains unclear whether greater dominance in English would lead Mandarin L2 learners of English to reduce their reliance on pitch when perceiving English lexical stress, given the high functional load of this cue in Mandarin. Moreover, research should determine whether the cue-weighting differences reflected in neural markers of phonetic processing are similarly modulated by language dominance (e.g., [87]), and the generalizability of language dominance effects should be tested in the perception of other types of contrasts (e.g., [12,26]). Finally, longitudinal studies are needed to determine whether changes in cue weighting co-vary with changes in language dominance over time, for example, in immigrants adapting to a new linguistic environment [104].

These findings also have practical implications. The observed differences in how bilinguals weight vowel quality, pitch, and duration cues indicate that speech perception and production patterns may diverge from monolingual norms without reflecting a speech or language impairment. Educators and clinicians should take language dominance into account when evaluating bilinguals, as it helps explain why certain cues are prioritized differently (cf. [102,103]). This understanding can guide targeted pronunciation or stress perception exercises, inform speech–language assessment and therapy, and support the development of instructional materials or auditory training programs that align with learners’ perceptual tendencies. Recognizing the role of dominance ensures that bilinguals are supported appropriately, without misinterpreting differences as deficits.

All in all, this study provides strong evidence that language dominance modulates perceptual attention in bilingual speech processing. Far from being fixed, cue-weighting strategies remain malleable, adapting to the demands of the dominant language. These findings reinforce the importance of treating dominance as a core variable in bilingualism research, contributing to a growing body of evidence that language experience can reshape speech perception mechanisms.

## Figures and Tables

**Figure 1 brainsci-15-01053-f001:**
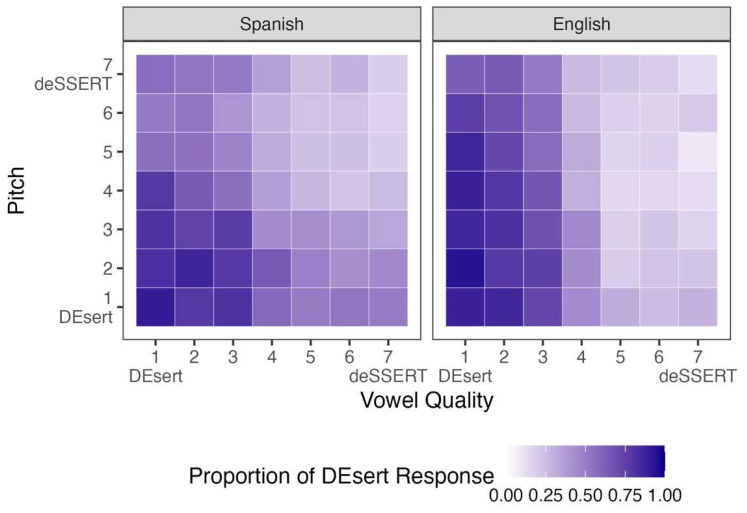
Participants’ proportions of *DEsert* selection when the stimuli varied by vowel quality and pitch.

**Figure 2 brainsci-15-01053-f002:**
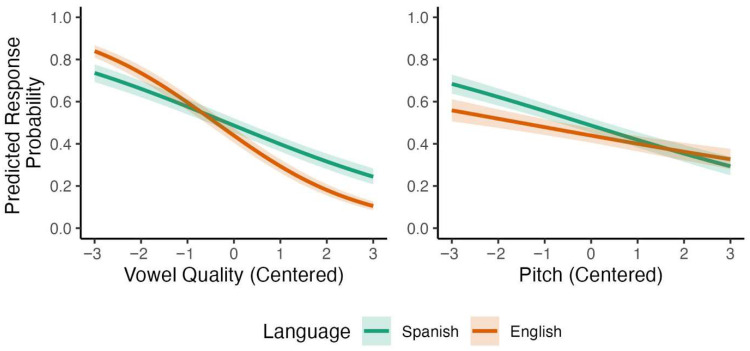
Predicted interactions with L1 for participants’ responses to stimuli varying by vowel quality and pitch.

**Figure 3 brainsci-15-01053-f003:**
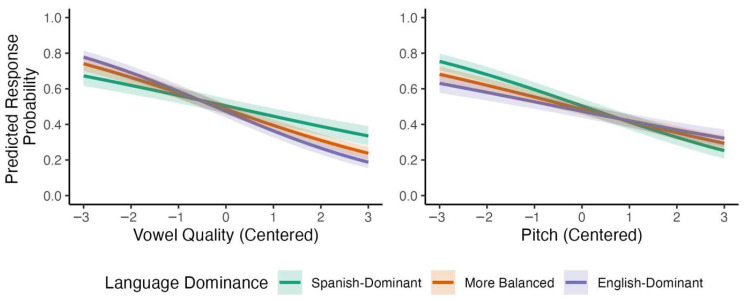
Predicted interactions with language dominance for bilinguals’ responses to stimuli varying by vowel quality and pitch.

**Figure 4 brainsci-15-01053-f004:**
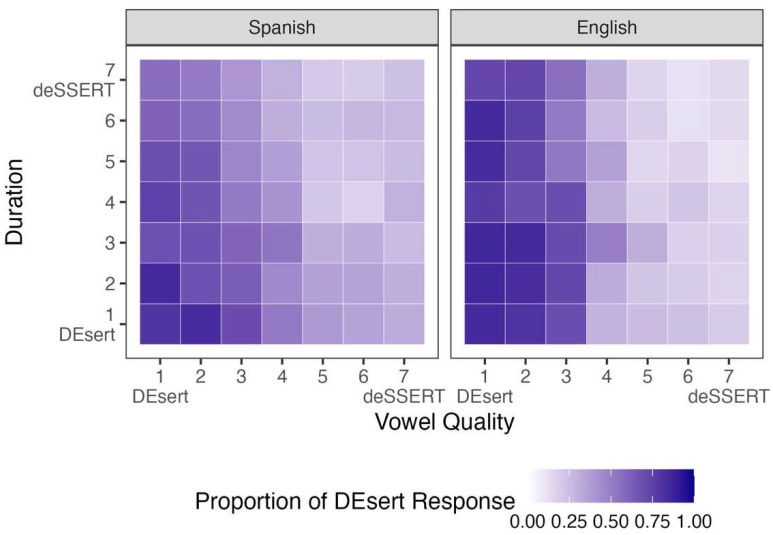
Participants’ proportions of *DEsert* selection when the stimuli varied by vowel quality and duration.

**Figure 5 brainsci-15-01053-f005:**
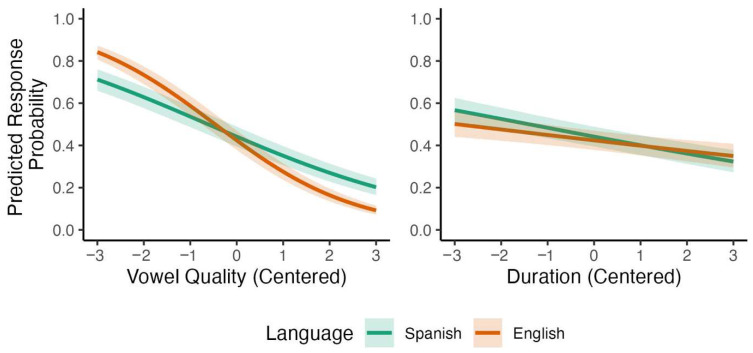
Predicted interactions with L1 for participants’ responses to stimuli varying by vowel quality and duration.

**Figure 6 brainsci-15-01053-f006:**
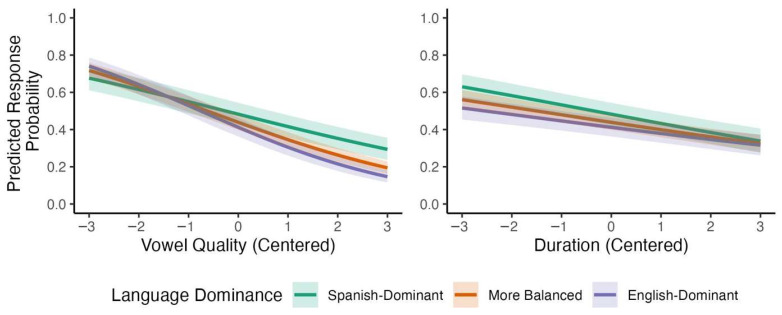
Predicted interactions with language dominance for bilinguals’ responses to stimuli varying by vowel quality and duration.

**Figure 7 brainsci-15-01053-f007:**
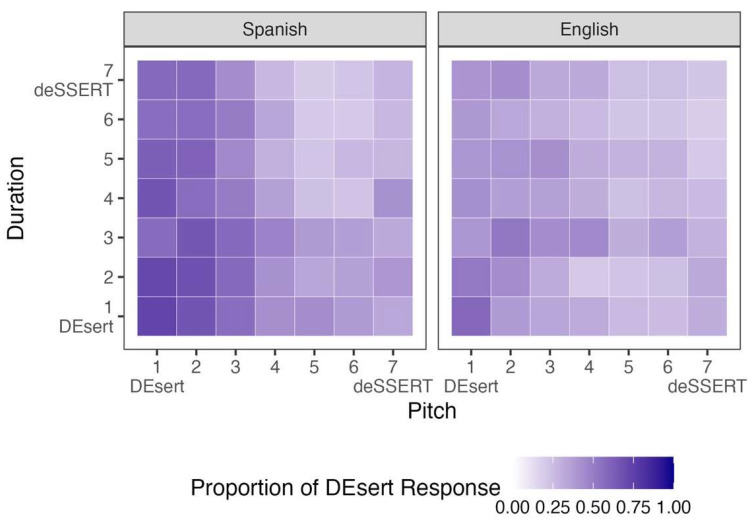
Participants’ proportions of *DEsert* selection when the stimuli varied by pitch and duration.

**Figure 8 brainsci-15-01053-f008:**
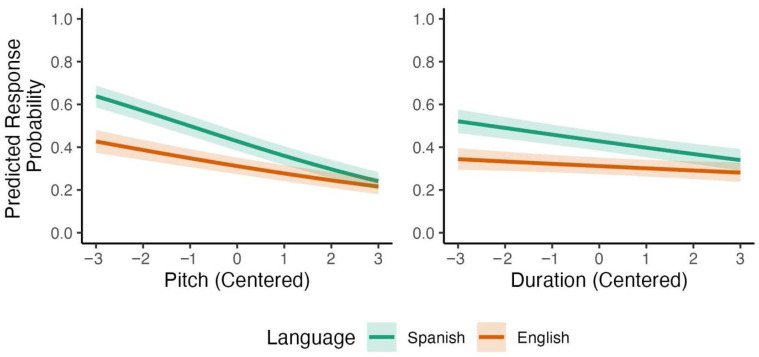
Predicted interactions with L1 for participants’ responses to stimuli varying by pitch and duration.

**Figure 9 brainsci-15-01053-f009:**
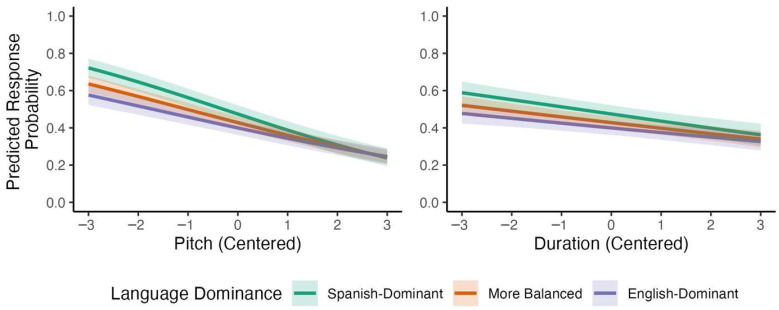
Predicted interactions with language dominance for bilinguals’ responses to stimuli varying by pitch and duration.

**Table 1 brainsci-15-01053-t001:** L1-Spanish L2-English Bilinguals’ Language Profile.

	Spanish	English
Age of onset of learning the language	0.2 (0.5, 0–2)	6.3 (5.1, 0–18)
Age of comfort using the language	1 (3.4, 0–20)	9 (6.1, 0–20)
Years of classes in the language	6.8 (6.0, 0–20)	12.1 (4.4, 3–19)
Years spent in Mexico (Spanish) or the US (English)	7.2 (8.1, 0–24)	13.7, 9.8, 0–39)
% use with friends	44.6 (32.6, 0–100)	55.4 (32.6, 0–100)
% use with family	67.4 (30.2, 0–100)	32.6 (30.2, 0–100)
% use at school/work	28.5 (22.8, 0–80)	71.5 (22.7, 20–100)
% use when you talk to yourself	43.8 (30.4, 0–100)	56.2 (30.4, 0–100)
% use when counting	44.9 (37.2, 0–100)	55.1 (37.2, 0–100)
Self-rated listening proficiency (/6)	5.3 (1.3, 1–6)	5.6 (0.6, 4–6)
Self-rated speaking proficiency (/6)	4.6 (1.6, 0–6)	5 (1, 3–6)
Self-rated reading proficiency (/6)	4.6 (1.6, 1–6)	5.5 (0.8, 3–6)
Self-rated writing proficiency (/6)	3.9 (1.8, 0–6)	5.2 (1.2, 1–6)
Feeling like oneself when speaking the language (/6)	4.8 (1.7, 0–6)	4.7 (1.8, 1–6)
Identification with language culture (/6)	4.8 (1.5, 0–6)	3.9 (1.9, 0–6)
Importance of using language like a native speaker (/6)	5 (1.5, 0–6)	4.7 (1.5, 0–6)
Desire to be perceived as a native speaker (/6)	4.1 (1.8, 0–6)	3.4 (2.2, 0–6)

Note. Mean (SD, range).

**Table 2 brainsci-15-01053-t002:** Acoustic Measurements of the Selected Naturally Produced Stimuli.

	*DEsert*		*deSSERT*	
	First Syllable	Second Syllable	First Syllable	Second Syllable
Duration (ms)	147	216	109	268
F0 (Hz)	275	208	227	223
Intensity (dB)	74	70	70	71
F1 (Hz)	669	556	432	597
F2 (Hz)	1845	1835	1841	1703
F3 (Hz)	2860	2242	2943	2082

Reproduced from [1], with permission of the Acoustical Society of America. Copyright [2025], Acoustical Society of America.

**Table 3 brainsci-15-01053-t003:** Results of Bayesian Logistic Regression on Participants’ Responses When Stimuli Varied by Vowel Quality and Pitch.

Parameter	Posterior Mean	95% CI (Lower)	95% CI (Higher)	*p* (*β* > 0)	*p* (*β* < 0)
Intercept	−0.053	−0.193	0.088	0.233	0.767
**Vowel Quality**	−0.36	−0.414	−0.308	<0.001	**>0.999**
**Pitch**	−0.275	−0.328	−0.223	<0.001	**>0.999**
**L1**	−0.193	−0.347	−0.038	0.007	**0.993**
Vowel Quality × Pitch	0.016	−0.011	0.043	0.882	0.118
**Vowel Quality × L1 (English)**	−0.274	−0.319	−0.228	<0.001	**>0.999**
**Pitch × L1 (English)**	0.116	0.073	0.159	**>0.999**	<0.001
Vowel Quality × Pitch × L1 (English)	0.01	−0.013	0.033	0.803	0.197

Note. Bolded fixed effects are credible.

**Table 4 brainsci-15-01053-t004:** Results of Bayesian Logistic Regression on Participants’ Responses When Stimuli Varied by Vowel Quality and Duration.

Parameter	Posterior Mean	95% CI (Lower)	95% CI (Higher)	*p* (*β* > 0)	*p* (*β* < 0)
Intercept	−0.227	−0.412	−0.04	0.009	0.991
**Vowel Quality**	−0.381	−0.432	−0.331	<0.001	**>0.999**
**Duration**	−0.169	−0.219	−0.118	<0.001	**>0.999**
L1	−0.076	−0.318	0.165	0.264	0.736
**Vowel Quality × Duration**	0.025	−0.001	0.05	**0.972**	0.028
**Vowel Quality × L1 (English)**	−0.281	−0.327	−0.234	<0.001	**>0.999**
**Duration × L1 (English)**	0.064	0.019	0.108	**0.998**	0.002
**Vowel Quality × Duration × L1 (English)**	−0.022	−0.045	0.001	0.029	**0.971**

Note. Bolded fixed effects are credible.

**Table 5 brainsci-15-01053-t005:** Results of Bayesian Logistic Regression on Participants’ Responses When Stimuli Varied by Pitch and Duration.

Parameter	Posterior Mean	95% CI (Lower)	95% CI (Higher)	*p* (*β* > 0)	*p* (*β* < 0)
Intercept	−0.291	−0.477	−0.108	0.001	0.999
**Pitch**	−0.287	−0.328	−0.244	<0.001	**>0.999**
**Duration**	−0.125	−0.166	−0.084	<0.001	**>0.999**
**L1**	−0.508	−0.752	−0.264	<0.001	**>0.999**
Pitch × Duration	−0.006	−0.027	0.015	0.282	0.718
**Pitch × L1 (English)**	0.121	0.081	0.161	**>0.999**	<0.001
**Duration × L1 (English)**	0.076	0.036	0.117	**>0.999**	<0.001
Pitch × Duration × L1 (English)	0.007	−0.013	0.027	0.750	0.250

Note. Bolded fixed effects are credible.

## Data Availability

The data upon which this manuscript is based were collected by Natalia Irene Minjárez-Oppenheimer for her M.A. Thesis at the University of Texas at El Paso (2024), a subset of which was reported in the thesis. Upon publication, the materials, data analysis and visualization scripts, and data will be made available under the corresponding Open Science Framework project (https://osf.io/t3fgp/).

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
