# Peer review of "Perceptual Plasticity in Bilinguals: Language Dominance Reshapes Acoustic Cue Weightings"

_brainsci, 2025, doi:10.3390/brainsci15101053_

Round 1
Reviewer 1 Report
Comments and Suggestions for Authors
This paper aims to assess the effect of language dominance on the weighting of attention to linguistic cues. The authors report that Spanish English bilinguals relied less on vowel quality and more on pitch and duration of vowel sounds. Further, that within the bilingual sample, English dominant bilinguals showed more of a reliance on vowel quality whereas Spanish dominant participants showed more of a reliance on pitch and duration. The authors conclude that both environment and language dominance continue to shape our linguistic attention allocation.
Overall, the paper is clear and, mostly, does an excellent job of reporting the methods, analysis and results (minor requests for more information below). However, I feel that between the introduction and discussion, I am still left with little confidence that the writer has a wider goal here. What is this contributing to? Exactly how does the data seen here (and not just thinking about levels of significance but also the small observable difference here really) have consequences for what we understand about language learning? Do you think these differences would be seen in combination with any other cues (given that language isn’t learned in isolation of other types of cue)? Or is the consequence in the contribution to plasticity theories? In which case, what does this change? I think the paper is currently a little unfinished in its current form for publication unfortunately and requires some deeper thinking to explore what could be much more interesting than is currently there.
Abstract, Highlights, and Introductions
1) Lines 53 to 60 outline that those who begin L2 language learning after childhood have more of an influence of the L1 language due to experience. I appreciate that you have pointed readers to a review here but it may be worth dedicating a few lines to show communicate that length of time in expereince with the L1 language is not the full picture. There are also factors related to brain development and whilst research shows that the brain retains plasticity – as do your findings here – it is still important not to over-simplify the factors in play in learning L2 after childhood compared with before.
2) Line 86: It is worth defining tasks and key concepts, such as “stop voicing contrasts”. At times it is written without a wider audience in mind.
3) Overall, I really enjoyed reading the introduction. I particularly enjoyed how clear the comparison paragraphs were between the use of the relevant cues in English and Spanish language. However, other than supporting the neurological work, I am still left unclear on why this matters. Does it refute other theories? Is this work to distinguish between theories and better our understanding? Or will this contribute to changes in our understanding of language teaching? As it stands, the introduction almost pitches the work as a very incremental build on the neurological work but I do think that it has more to contribute than this.
Methods
4) For the participants section, the extensive description of the bilingual group is really ncie and very interesting. There is a large dominance of female participants. Is this addressed in the discussion?
5) This study was conducted as part of a larger study with other tasks. Could more information be given in the methods as to how that was done? Was the order of tasks always the same or randomised? Where in the running order did this study come?
Results
6) For the language dominance aspect of data analysis, there could be more information in relation to the creation of the three groups. How many are in each group? What’s the variance within these groups?
Discussion
7) I can see the changes in data reported for the English vs bilingual pps, alongside the switch in attention around the +1-2 mark for the three levels of dominance within the bilingual group. These appear to be very small differences though on both sides of the switch and I wonder if this were to be conducted outside of isolation, in the context of a more complex cue environment, whether these differences would be observed?
8) For lines 535 to 537. I like that it is becoming clear that this data challenges theoretical accounts. These are not referenced and haven’t yet been outlined. It would be good to give more space to this so the consequences of the two opposing types of theories are clear.
9) There does not appear to be any sort of section that contextualises results with limitations – for example, how is the large imbalance of females, small sample size and wide variance across the small sample size likely to have affected results?
10) The discussion overall does a good job of summarising research. It offers some further incremental aspects for future research. But there could be more space here for really contextualising what this means. Why does this matter? What is the overall objective here?
Author Response
Response to Review #1
General Comment: Overall, the paper is clear and, mostly, does an excellent job of reporting the methods, analysis and results (minor requests for more information below). However, I feel that between the introduction and discussion, I am still left with little confidence that the writer has a wider goal here. What is this contributing to? Exactly how does the data seen here (and not just thinking about levels of significance but also the small observable difference here really) have consequences for what we understand about language learning? Do you think these differences would be seen in combination with any other cues (given that language isn’t learned in isolation of other types of cue)? Or is the consequence in the contribution to plasticity theories? In which case, what does this change? I think the paper is currently a little unfinished in its current form for publication unfortunately and requires some deeper thinking to explore what could be much more interesting than is currently there.
General Response: We thank the reviewer for this important comment. We added a paragraph to the introduction to highlight the broader picture within which this study can be situated: “Addressing this question is critical not only for testing the limits of the Cue-Weighting Transfer Hypothesis but also for advancing broader theories of bilingual speech perception, which must account for how cue-weighting routines evolve with shifting patterns of language dominance. Additionally, this research is important for broader theories of bilingual language learning and plasticity. Demonstrating that bilinguals can adjust reliance on L1-preferred cues in favor of L2-preferred ones would suggest that the perceptual system remains dynamically reconfigurable well into adulthood, rather than being fixed by early experience. It would also clarify how listeners integrate multiple cues in real-world learning contexts, where speech is rarely processed in isolation from other dimensions. More broadly, such findings would contribute to ongoing debates about the limits of perceptual flexibility in L2 acquisition and provide a framework for distinguishing systematic bilingual adaptations from impaired processing.”
We also added a paragraph to the discussion to highlight the theoretical importance of our results for theories of plasticity: “Beyond replicating and extending prior work, the present findings have broader consequences for how we conceptualize language learning and plasticity. The dynamic cue reweighting observed here demonstrates that perceptual attention to prosodic cues is not rigidly constrained by the L1 but instead reflects ongoing adaptation to patterns of input across both languages. This provides behavioral evidence for plasticity in bilingual speech perception, showing that the weighting of prosodic dimensions can be recalibrated with sufficient L2 experience. Although our study examined cues in isolation, the results have clear implications for cue integration: If bilinguals shift their reliance away from pitch and duration and toward vowel quality when processing English stress, similar shifts are likely to shape how they integrate multiple cues in more naturalistic contexts. While the dominance-related differences we observed were relatively small in magnitude, their systematicity suggests genuine underlying perceptual adjustments. Importantly, even subtle shifts in cue weighting can have cascading effects in speech processing, potentially delaying lexical access if more competitors are activated when the cues are not used efficiently. Future work will need to test whether the shifts we observed in the current study extend to complex, multi-cue environments such as continuous speech. The present findings therefore contribute to a broader understanding of bilingual learning, supporting models in which perceptual systems flexibly redistribute attention across cues as a function of language experience. They also underscore that bilingual speech patterns, often described as “non-native”, should not be taken as evidence of impairment but instead as evidence of systematic adaptations to bilingual input (Bayram et al., 2021; Kutlu et al., 2022).”
We are thankful to the reviewer for this comment, as these additions significantly strengthen the paper.
Comment 1: Lines 53 to 60 outline that those who begin L2 language learning after childhood have more of an influence of the L1 language due to experience. I appreciate that you have pointed readers to a review here but it may be worth dedicating a few lines to show communicate that length of time in experience with the L1 language is not the full picture. There are also factors related to brain development and whilst research shows that the brain retains plasticity – as do your findings here – it is still important not to over-simplify the factors in play in learning L2 after childhood compared with before.
Response 1: We thank the reviewer for this valuable comment. We added a sentence acknowledging that neurodevelopmental effects may also be at play but recognizing that such effects are inconclusive: “These age-of-acquisition effects may reflect several factors, including longer experience with the L1 before exposure to the L2 and possible changes in neurocognitive development. Evidence for strict maturational constraints in speech perception, however, is inconclusive; rather, research suggests that age-related changes may be driven by L1 experience and interact with other factors such as L2 proficiency and attention control (e.g., Li et al., 2025).
Comment 2: Line 86: It is worth defining tasks and key concepts, such as “stop voicing contrasts”. At times it is written without a wider audience in mind.
Response 2: We thank the reviewer for this comment. We added examples to clarify what “stop voicing contrasts” mean.
Comment 3: Overall, I really enjoyed reading the introduction. I particularly enjoyed how clear the comparison paragraphs were between the use of the relevant cues in English and Spanish language. However, other than supporting the neurological work, I am still left unclear on why this matters. Does it refute other theories? Is this work to distinguish between theories and better our understanding? Or will this contribute to changes in our understanding of language teaching? As it stands, the introduction almost pitches the work as a very incremental build on the neurological work but I do think that it has more to contribute than this.
Response 3: We thank the reviewer for this valuable comment. We added a statement after the paragraph introducing the research questions that clarifies the theoretical contribution of the study: “Addressing this question is critical not only for testing the limits of the Cue-Weighting Transfer Hypothesis but also for advancing broader theories of bilingual speech perception, which must account for how cue-weighting routines evolve with shifting patterns of language dominance.” As indicated above (see general response), we also added a paragraph in the introduction clarifying the broader significance of this research: “This question matters not only for models of cue weighting, but also for broader theories of bilingual language learning and plasticity. Demonstrating that bilinguals can adjust reliance on L1-preferred cues in favor of L2-preferred ones would suggest that the perceptual system remains dynamically reconfigurable well into adulthood, rather than being fixed by early experience. It would also clarify how listeners integrate multiple cues in real-world learning contexts, where speech is rarely processed in isolation from other dimensions. More broadly, such findings would contribute to ongoing debates about the limits of perceptual flexibility in L2 acquisition and provide a framework for distinguishing systematic bilingual adaptations from impaired processing.”
Comment 4: For the participants section, the extensive description of the bilingual group is really nice and very interesting. There is a large dominance of female participants. Is this addressed in the discussion?
Response 4: We now acknowledge the limitations of our study in the discussion section: “Despite the robustness of these findings, several limitations should be acknowledged. First, the participant pool was relatively small and narrow, consisting only of 39 highly proficient Spanish-English bilinguals and English monolinguals, most of whom were female. Our design allowed us to isolate the role of dominance while controlling for proficiency and age of acquisition of Spanish, but it also limits the generalizability of the results to bilinguals with other L1-L2 pairings and to male bilinguals (though gender has not been reported to influence cue weightings in speech perception).”
Comment 5: This study was conducted as part of a larger study with other tasks. Could more information be given in the methods as to how that was done? Was the order of tasks always the same or randomised? Where in the running order did this study come?
Response 5: Participants completed this cue weighting experiment after they completed a visual-world eye-tracking experiment. We have added this clarification in Footnote 4.
Comment 6: For the language dominance aspect of data analysis, there could be more information in relation to the creation of the three groups. How many are in each group? What’s the variance within these groups?
Response 6: We thank the reviewer for this comment. We did not conduct group analyses; language dominance was always a continuous variable in the models, and the sub-group division was made strictly for visualization purposes. We realize now that the method section did not make that clear. We clarified this information in the method section and reiterated it when presenting the first set of results with language dominance as a factor. “Recall that, in these analyses, language dominance is entered in the models as a continuous variable, and participants are divided into terciles only for visualization purposes.”
Comment 7: I can see the changes in data reported for the English vs bilingual pps, alongside the switch in attention around the +1-2 mark for the three levels of dominance within the bilingual group. These appear to be very small differences though on both sides of the switch and I wonder if this were to be conducted outside of isolation, in the context of a more complex cue environment, whether these differences would be observed?
Response 7: We acknowledge the reviewer’s observation that the differences in cue weighting across groups, while systematic, are relatively small in magnitude. We view these differences as theoretically meaningful, however, as they align with predictions of the Cue-Weighting Transfer Hypothesis and reveal a clear modulation of reliance on pitch and duration cues as a function of bilinguals’ language dominance. Even subtle shifts in cue weighting can have cascading effects in speech processing, potentially delaying lexical access if more competitors are activated when the cues are not used efficiently. At the same time, we agree that future research should examine whether such shifts are maintained in more naturalistic listening conditions where multiple cues co-occur and interact, something we now mention in our discussion section: “Although our study examined cues in isolation, the results have clear implications for cue integration: If bilinguals shift their reliance away from pitch and duration and toward vowel quality when processing English stress, similar shifts are likely to shape how they integrate multiple cues in more naturalistic contexts. While the dominance-related differences we observed were relatively small in magnitude, their systematicity suggests genuine underlying perceptual adjustments. Importantly, even subtle shifts in cue weighting can have cascading effects in speech processing, potentially delaying lexical access if more competitors are activated when the cues are not used efficiently. Future work will need to test whether the shifts we observed in the current study extend to complex, multi-cue environments such as continuous speech. The present findings therefore contribute to a broader understanding of bilingual learning, supporting models in which perceptual systems flexibly redistribute attention across cues as a function of language experience. They also underscore that bilingual speech patterns, often described as “non-native”, should not be taken as evidence of impairment but instead as evidence of systematic adaptations to bilingual input (Bayram et al., 2021; Kutlu et al., 2022).”
Comment 8: For lines 535 to 537. I like that it is becoming clear that this data challenges theoretical accounts. These are not referenced and haven’t yet been outlined. It would be good to give more space to this so the consequences of the two opposing types of theories are clear.
Response 8: We thank the reviewer for pointing this out. We have now cited Dupoux and colleagues’ “stress deafness” account, which was the theoretical position we had in mind. At the same time, our study was not designed to directly adjudicate between stress deafness and alternative accounts, since such a test would require using comparable paradigms. Instead, our focus was on how cue weighting varies as a function of language dominance and the gradient contribution of specific acoustic cues, which motivated a different experimental design. For this reason, rather than positioning our study as a direct challenge to stress deafness, we frame our contribution as complementary: It highlights the importance of considering bilingual experience and cue-specific reweighting, which extends the scope of current theoretical debates.
Comment 9: There does not appear to be any sort of section that contextualises results with limitations – for example, how is the large imbalance of females, small sample size and wide variance across the small sample size likely to have affected results?
Response 9: We thank the reviewer for this comment. We now acknowledge the generalizability limitations of our study in the discussion section: “Despite the robustness of these findings, several limitations should be acknowledged. First, the participant pool was relatively small and narrow, consisting only of 39 highly proficient Spanish-English bilinguals and English monolinguals, most of whom were female. Our design allowed us to isolate the role of dominance while controlling for proficiency and age of acquisition of Spanish, but it also limits the generalizability of the results to bilinguals with other L1-L2 pairings and to male bilinguals (though gender has not been reported to influence cue weightings in speech perception).”
Comment 10: The discussion overall does a good job of summarising research. It offers some further incremental aspects for future research. But there could be more space here for really contextualising what this means. Why does this matter? What is the overall objective here?
Response 10: We thank the reviewer for this valuable comment. We added a paragraph about practical implications, which we hope strengthens the paper by clarifying why this research matters: “These findings also have practical implications. The observed differences in how bilinguals weight vowel quality, pitch, and duration cues indicate that speech perception and production patterns may diverge from monolingual norms without reflecting a speech or language impairment. Educators and clinicians should take language dominance into account when evaluating bilinguals, as it helps explain why certain cues are prioritized differently (cf. Bayram et al., 2021; Kutlu et al., 2022). This understanding can guide targeted pronunciation or stress perception exercises, inform speech-language assessment and therapy, and support the development of instructional materials or auditory training programs that align with learners’ perceptual tendencies. Recognizing the role of dominance ensures that bilinguals are supported appropriately, without misinterpreting differences as deficits.”
Reviewer 2 Report
Comments and Suggestions for Authors
Dear authors, thank you for sharing your study. It is indeed a sound insight into the aspects of language dominance in bilinguals' speech perception. I would just recommend adding or emphasing the practical implications of your study, e.g. in regards to bilingualism, language acquisition, or sociolinguistics. You may also want to do some quick editing for minor typos/spelling errors as “nativespeaker” (line 248, p. 5).
Author Response
Response to Reviewer #2
Comment 1: Dear authors, thank you for sharing your study. It is indeed a sound insight into the aspects of language dominance in bilinguals’ speech perception. I would just recommend adding or emphasing the practical implications of your study, e.g. in regards to bilingualism, language acquisition, or sociolinguistics. You may also want to do some quick editing for minor typos/spelling errors as “nativespeaker” (line 248, p. 5).
Response 1: We thank the reviewer for this comment. We added an implication section to the discussion section: “These findings also have practical implications. The observed differences in how bilinguals weight vowel quality, pitch, and duration cues indicate that speech perception and production patterns may diverge from monolingual norms without reflecting a speech or language impairment. Educators and clinicians should take language dominance into account when evaluating bilinguals, as it helps explain why certain cues are prioritized differently (cf. Bayram et al., 2021; Kutlu et al., 2022). This understanding can guide targeted pronunciation or stress perception exercises, inform speech-language assessment and therapy, and support the development of instructional materials or auditory training programs that align with learners’ perceptual tendencies. Recognizing the role of dominance ensures that bilinguals are supported appropriately, without misinterpreting differences as deficits.” We also searched for the “nativespeaker” typo but did not find it.
Reviewer 3 Report
Comments and Suggestions for Authors
The present study explored how language dominance influenced selective attention to acoustic cues in L2. The results showed that English dominant speakers showed likeliness of the English L1 speakers, showing a reliance on vowel quality and decreased reliance on pitch and duration. By contrast, the Spanish dominant speakers showed increased reliance on pitch and duration. The results support a model of experience-based speech perception model. Overall the present study is well-conducted and has some contribution to this field. There are some issues worth further attention.
- The sample size should be determined by power analysis, especially for the group of different language dominance. Between-group design usually calls for a large sample size.
- For the materials, how about the word frequency and word length of the words in different conditions?
- For Figure 3. English dominant, a “t”is missing.
- The discussion is relatively short. The authors should extend the discussion by explaining the detailed results. They provided very detailed results and the results should be discussed with previous studies and explain them in depth.
- The authors mentioned tonal languages, such as Mandarin and Cantonese. How different pitch variance and complexity could be shaped by the language dominance and their findings. The authors could provide the possible hypotheses.
Author Response
Response to Report #3
Comment 1: The sample size should be determined by power analysis, especially for the group of different language dominance. Between-group design usually calls for a large sample size.
Response 1: We thank the reviewer for this comment. We did not conduct group analyses; language dominance was always a continuous variable in the models, and the sub-group division was made strictly for visualization purposes. We realize now that the method section did not make that clear. We clarified this information in the method section and reiterated it when presenting the first set of results with language dominance as a factor. “Recall that, in these analyses, language dominance is entered in the models as a continuous variable, and participants are divided into terciles only for visualization purposes.” We did not conduct a power analysis because this is not yet an expectation for studies conducted in the field of Linguistics. That being said, it is something we intend to do in the future. We could conduct such an analysis now, but we are not sure there would be any point in doing so retroactively.
Comment 2: For the materials, how about the word frequency and word length of the words in different conditions?
Response 2: Participants heard only DEsert and deSSERT throughout the experiment. Both are frequent English words, and small frequency differences between the two words should be attenuated by the fact that the participants only heard these two words throughout the experiment. The word duration varied as a function of the duration manipulation—words at Step 1 of duration had the duration of DEsert and words at step 7 of duration had the duration of deSSERT. The duration values are provided in Table 2 (DEsert = 363 ms, deSSERT = 377).
Comment 3: For Figure 3. English dominant, a “t”is missing.
Response 3: We thank the reviewer for noticing this. We have regenerated the figures, which were just a bit too small for all the text. The new figures have been inserted in the paper.
Comment 4: The discussion is relatively short. The authors should extend the discussion by explaining the detailed results. They provided very detailed results and the results should be discussed with previous studies and explain them in depth.
Response 4: We have lengthened the discussion section by providing a more thorough discussion of our findings, by acknowledging the limitations of the study, and by adding a practical implications section.
Comment 5: The authors mentioned tonal languages, such as Mandarin and Cantonese. How different pitch variance and complexity could be shaped by the language dominance and their findings. The authors could provide the possible hypotheses.
Response 5:We clarified one question that future research could seek to answer: “Speakers of tonal languages such as Mandarin are known to transfer their use of pitch to the perception of lexical stress in English, a finding attributed to the high functional load of pitch from lexical tones in the L1 (e.g., Choi, 2022; Choi et al., 2019; Qin et al., 2017; Wang et al., 2024). However, it remains unclear whether greater dominance in English would lead Mandarin L2 learners of English to reduce their reliance on pitch when perceiving English lexical stress given the very high functional load of this cue in Mandarin.”
Reviewer 4 Report
Comments and Suggestions for Authors
I enjoyed reading the paper. It is well-written. However, I suggest some minor corrections:
- The description of the Bilingual Language Profile modifications would benefit from clearer justification, particularly regarding the exclusion of certain items.
- Claims such as “this is the first study” should be tempered for caution (e.g., “to our knowledge, this is among the first…”).
- The acknowledgment of ChatGPT use should be phrased with more academic caution to reassure readers that the substantive analysis and interpretation remain the authors’ responsibility.
- Technical details in the stimuli creation process may be briefly simplified for readers outside experimental phonetics.
- The limitations section could be expanded slightly to acknowledge the relatively narrow participant pool and the reliance on self-reported dominance measures.
- The research questions and hypotheses are clear, but they could be explicitly stated as bullet points or numbered aims at the end of the introduction to guide the reader more directly.
- Since “language dominance” is a multifaceted construct, it may help to remind readers in the discussion that results reflect composite scores rather than single objective measures of proficiency.
- The bilingual sample (n = 39) is relatively modest, and dominance subgroups (Spanish-dominant, balanced, English-dominant) may be underpowered. The authors should acknowledge this more explicitly as a limitation when interpreting within-group effects.
- The stimuli were well controlled, but the description of the acoustic manipulations (linear interpolation in Praat) could be clarified for less technical readers. For instance, specifying how vowel quality was interpolated across formants would improve transparency.
- The choice of Bayesian logistic regression is justified, but the rationale could be briefly explained in accessible terms (e.g., why Bayesian inference was preferable to frequentist approaches for this dataset).
- The future directions section is strong, but adding a brief practical implication (e.g., for bilingual education, speech therapy, or assessment) would broaden the manuscript’s impact beyond theoretical debates.
Author Response
Response to Reviewer #4
Comment 1: The description of the Bilingual Language Profile modifications would benefit from clearer justification, particularly regarding the exclusion of certain items.
Response 1: We thank the reviewer for this comment. We added a footnote to further justify the need to modify the questionnaire: “The questionnaire was modified because some of the questions from the original tool do not reflect the reality of Spanish-English bilinguals who live on the U.S.-Mexico border. For example, participants whose country of residence is Mexico may still spend a significant amount of time in the US if they are border commuters. The questionnaire was modified to reflect this reality.” The exclusion of certain items is motivated in the main text: “… as the remaining questions were adapted and no longer relativistic.” Because the questions were no longer relativistic, they would not speak about language dominance, which is a relativistic concept.
Comment 2: Claims such as “this is the first study” should be tempered for caution (e.g., “to our knowledge, this is among the first…”).
Response 2: We thank the reviewer for bringing this to our attention. We revised the text accordingly.
Comment 3: The acknowledgment of ChatGPT use should be phrased with more academic caution to reassure readers that the substantive analysis and interpretation remain the authors’ responsibility.
Response 3: We revised the text to make this clearer: “During the preparation of this manuscript, the authors used ChatGPT for the purpose of improving the data analysis and visualization scripts and to increase the clarity of the writing. The tool was not involved in the design of the study, nor did it directly handle the analysis or interpretation of the results. All outputs were critically reviewed and edited by the authors, who take full responsibility for the content of this publication.”
Comment 4: Technical details in the stimuli creation process may be briefly simplified for readers outside experimental phonetics
Response 4: We condensed and simplified the description of the stimuli.
Comment 5: The limitations section could be expanded slightly to acknowledge the relatively narrow participant pool and the reliance on self-reported dominance measures.
Response 5: We thank the reviewer for this comment. We have revised the discussion section to better acknowledge the limitations of our study: “Despite the robustness of these findings, several limitations should be acknowledged. First, the participant pool was relatively small and narrow, consisting only of 39 highly proficient Spanish-English bilinguals and English monolinguals, most of whom were female. Our design allowed us to isolate the role of dominance while controlling for proficiency and age of acquisition of Spanish, but it also limits the generalizability of the results to bilinguals with other L1-L2 pairings and possibly to male bilinguals (though gender has not been reported to influence cue weightings in speech perception). Second, language dominance was assessed only through the self-report measures provided by the Bilingual Language Profile (Birdsong et al., 2012; Gertken et al., 2014). Although widely used and informative, such measures may not provide the same degree of precision or accuracy as objective behavioral measures.”
Comment 6: The research questions and hypotheses are clear, but they could be explicitly stated as bullet points or numbered aims at the end of the introduction to guide the reader more directly.
Response 6: We appreciate this comment from the reviewer. We added our aims in bulleted form to the paper and agree that it strengthens its focus: “Building on this body of research, and considering the cross-linguistic differences in how lexical stress is cued in English and Spanish, the present study was designed with two main aims: (1) To determine whether Spanish-English bilinguals and English monolinguals differ in their reliance on vowel quality, pitch, and duration when perceiving English lexical stress. (2) To test whether language dominance predicts individual differences in bilinguals’ cue weighting, with the expectation that English-dominant bilinguals will show greater reliance on vowel quality and reduced reliance on pitch and duration compared to Spanish-dominant bilinguals.” We also revised our first discussion paragraph to connect back to these two aims: “The present study addressed two main questions: (1) whether Spanish-English bilinguals differ from English monolinguals in their reliance on vowel quality, pitch, and duration when perceiving English lexical stress, and (2) whether language dominance predicts individual differences in cue weighting among highly proficient bilinguals. Consistent with our first aim, the results revealed clear cross-linguistic differences: Compared to monolingual English listeners, Spanish-English bilinguals relied less on vowel quality and more on pitch and duration when identifying stress in English words. These differences reflect the influence of L1 experience on cue weighting, supporting the notion that selective attention to acoustic dimensions is language-specific. Critically, in line with our second aim, language dominance emerged as a credible predictor of cue weighting within the bilingual group…”
Comment 7: Since “language dominance” is a multifaceted construct, it may help to remind readers in the discussion that results reflect composite scores rather than single objective measures of proficiency.
Response 7: We thank the reviewer for this comment. We added a sentence at the end of our first discussion paragraph to highlight this: “These results should be interpreted as reflecting relative dominance across multiple dimensions rather than a single objective metric, underscoring the multifaceted nature of this construct.”
Comment 8: The bilingual sample (n = 39) is relatively modest, and dominance subgroups (Spanish-dominant, balanced, English-dominant) may be underpowered. The authors should acknowledge this more explicitly as a limitation when interpreting within-group effects.
Response 8: We now acknowledge the limited sample size in the discussion section (see Response 5). We did not conduct group analyses; language dominance was always a continuous variable in the models, and the sub-group division was made strictly for visualization purposes. We realize now that the method section did not make that clear. We clarified this information and reiterated it when presenting the first set of results with language dominance as a factor. “Recall that, in these analyses, language dominance is entered in the models as a continuous variable, and participants are divided into terciles only for visualization purposes.”
Comment 9: The stimuli were well controlled, but the description of the acoustic manipulations (linear interpolation in Praat) could be clarified for less technical readers. For instance, specifying how vowel quality was interpolated across formants would improve transparency.
Response 9: It is difficult to both simplify this information (Comment 4) and provide additional details about the acoustic manipulations. We simplified the information to make it more accessible to the reader but clarified that vowels had their formant frequencies and bandwidth manipulated: “The stimuli were manipulated in Praat (version 6.0.46; Boersma & Weenink, 2019) to vary along three acoustic dimensions that signal lexical stress: vowel quality, pitch, and duration. Vowel quality was adjusted by gradually transforming the spectral characteristics (i.e., first, second, and third formant frequencies and bandwidths) of the vowels from DEsert to deSSERT in seven evenly spaced steps, producing a continuum from one vowel realization to the other. Pitch and duration were similarly varied in seven steps between the two endpoint words. For each stimulus, two dimensions were varied independently while the third was held at a neutral midpoint, resulting in three sets of stimuli: vowel quality × pitch, vowel quality × duration, and pitch × duration. Overall intensity was neutralized across the two syllables and normalized across items.”
Comment 10: The choice of Bayesian logistic regression is justified, but the rationale could be briefly explained in accessible terms (e.g., why Bayesian inference was preferable to frequentist approaches for this dataset).
Response 10: We added the following justification: “Bayesian modeling was chosen because it provides full posterior distributions, allowing probabilistic statements about the credibility of effects (e.g., P(β > 0)), and handles continuous predictors and complex hierarchical structures more robustly than frequentist models, which sometimes exhibited convergence issues. Bayesian methods also naturally accommodate uncertainty in parameter estimates and facilitate visualization of marginal effects with credible intervals, which is especially useful for interpreting interactions with continuous predictors.”
Comment 11: The future directions section is strong, but adding a brief practical implication (e.g., for bilingual education, speech therapy, or assessment) would broaden the manuscript’s impact beyond theoretical debates.
Response 11: We thank the reviewer for this important point. We added the following paragraph to our discussion section: “These findings also have practical implications. The observed differences in how bilinguals weight vowel quality, pitch, and duration cues indicate that speech perception and production patterns may diverge from monolingual norms without reflecting a speech or language impairment. Educators and clinicians should take language dominance into account when evaluating bilinguals, as it helps explain why certain cues are prioritized differently (cf. Bayram et al., 2021; Kutlu et al., 2022). This understanding can guide targeted pronunciation or stress perception exercises, inform speech-language assessment and therapy, and support the development of instructional materials or auditory training programs that align with learners’ perceptual tendencies. Recognizing the role of dominance ensures that bilinguals are supported appropriately, without misinterpreting differences as deficits.”
Round 2
Reviewer 3 Report
Comments and Suggestions for Authors
I think the authors addressed most of my concerns, but the layout should be changed to the journal format.